# Solvent Dependence of the Rheological Properties in Hydrogel Magnetorheological Plastomer

**DOI:** 10.3390/ijms21051793

**Published:** 2020-03-05

**Authors:** Norhiwani Mohd Hapipi, Saiful Amri Mazlan, U. Ubaidillah, Siti Aishah Abdul Aziz, Muntaz Hana Ahmad Khairi, Nur Azmah Nordin, Nurhazimah Nazmi

**Affiliations:** 1Engineering Materials and Structures (eMast) iKohza, Malaysia-Japan International Institute of Technology, Universiti Teknologi Malaysia, Jalan Sultan Yahya Petra, Kuala Lumpur 54100, Malaysia; hiwani87@gmail.com (N.M.H.); aishah118@gmail.com (S.A.A.A.); hana5700@gmail.com (M.H.A.K.); nurazmah.nordin@utm.my (N.A.N.); nurhazimah@utm.my (N.N.); 2Mechanical Engineering Department, Faculty of Engineering, Universitas Sebelas Maret, Jl. Ir. Sutami 36A Kentingan Jebres, Surakarta 57126, Indonesia; 3National Center for Sustainable Transportation Technology (NCSTT), Bandung 40132, Indonesia

**Keywords:** magnetorheological plastomer, polyvinyl alcohol, dimethyl sulfoxide, corrosion, hydrogel

## Abstract

Chemically crosslinked hydrogel magnetorheological (MR) plastomer (MRP) embedded with carbonyl iron particles (CIPs) exhibits excellent magnetic performance (MR effect) in the presence of external stimuli especially magnetic field. However, oxidation and desiccation in hydrogel MRP due to a large amount of water content as a dispersing phase would limit its usage for long-term applications, especially in industrial engineering. In this study, different solvents such as dimethyl sulfoxide (DMSO) are also used to prepare polyvinyl alcohol (PVA) hydrogel MRP. Thus, to understand the dynamic viscoelastic properties of hydrogel MRP, three different samples with different solvents: water, DMSO, and their binary mixtures (DMSO/water) were prepared and systematically carried out using the oscillatory shear. The outcomes demonstrate that the PVA hydrogel MRP prepared from precursor gel with water shows the highest MR effect of 15,544% among the PVA hydrogel MRPs. However, the samples exhibit less stability and tend to oxidise after a month. Meanwhile, the samples with binary mixtures (DMSO/water) show an acceptable MR effect of 11,024% with good stability and no CIPs oxidation. Otherwise, the sample with DMSO has the lowest MR effect of 7049% and less stable compared to the binary solvent samples. This confirms that the utilisation of DMSO as a new solvent affects the rheological properties and stability of the samples.

## 1. Introduction

Magnetorheological (MR) material is a material that consists of magnetic particles embedded in a carrier matrix. It has the capability of changing its properties by the application of external stimuli like a magnetic field due to the existence of magnetic particles. The MR material can be grouped into several types according to the difference of its carrier matrices, such as MR fluid (MRF), MR elastomer (MRE), MR foam, and MR gel (MRG). As the first developed MR material, MRF has been widely utilised in industrial applications such as vibration dissipation, damping, artificial muscles, drug delivery, valves, and actuators [1,2,3,4]. However, MRF exhibits several drawbacks, such as sedimentation and leaking problems that limit its application [5]. Thus, by replacing the fluid medium with a solid matrix, which is best known as MRE, the problems in MRF can be hindered [6]. Unlike the MRF, the magnetic particles inside MRE remain trapped in the matrix resulting in more stable mechanical properties of MRE. Unfortunately, MRE also has some drawback such as weak MR effects due to its hard nature of polymer matrix [7]. Therefore, a new class of MR material known as MR plastomer (MRP) has been developed recently due to its high MR effect and stability. As a new kind of intelligent MR material, MRP is prepared by dispersing micron-sized magnetic responsive particles into a low cross-linking polymer gel that has plastic characteristics such as hydrogel, polyurethane, paraffin, and polymer gel [8,9,10,11]. By referring to the high viscosity of polymer matrix, MRP becomes a promising candidate to substitute traditional MRF in some applications because no sedimentation of magnetic particles occurred [12]. Additionally, the magnetic particles inside the low cross-linked polymer matrix are moveable and not permanently trapped with the presence of an external magnetic field, which leads to a high MR effect in MRP.

At the early stage, the hydrogel matrix has been widely introduced as a carrying matrix to prepare MRP [7]. In general, hydrogel MRP comprises magnetic particles embedded inside a polymer matrix network swollen by a liquid solution like water and water-miscible organic solvent [13,14]. Hydrogel networks mainly include carrageenan, polysaccharides, agar, guar gum, and polymer derivatives such as polyvinyl alcohol (PVA). The MR behaviour of hydrogel MRP has been investigated extensively over the past decade and proven that the type of matrix strongly influences the mechanical properties of MRP. For instance, Mitsumata et al. [15,16], investigated the magnetic response of carrageenan magnetic hydrogel and found that the hydrogel exhibits negative MR effect. Moreover, the magnetic carrageenan gels display high reduction in storage modulus (G′) of ~107 Pa upon magnetisation. Meanwhile, Negami et al. [17] reported that the magnetic hydrogel using PVA as a matrix produced higher mechanical toughness compared to the carrageenan hydrogel. The yield point of magnetic PVA hydrogel was higher compared to that of the magnetic carrageenan with strain of 0.8 and 0.35, respectively. The result shows that the magnetic PVA hydrogel possesses better mechanical strength compared to the magnetic carrageenan hydrogel. The giant MR behaviour and high yield point made the PVA hydrogel is suitable to be used as damping materials, sensors, and smart actuators.

PVA is a synthetic hydrophilic polymer, which is mechanically soft, biocompatible, high tensile modulus, high abrasion resistance, and high strength in compression that can be chemically and physically crosslinked [18,19,20]. Wu et al. [18] studied physically cross-linked PVA hydrogel MRP using the freezing–thawing method. The gelation of PVA hydrogel MRP arises through the formation of hydrogen bonding by crystallisation. They found that the PVA hydrogel MRP is a promising candidate for the manufacture of MR material as its tensile strength value was close to that of the natural rubber. The highest MR effect achieved by the sample was 230%. Park et al. [20] compared the actuator behaviour of both chemically and physically cross-linked PVA hydrogel MRPs. The outcome demonstrated that the chemically cross-linked PVA hydrogel MRP exhibited lower stiffness and larger strains than those of the physically cross-linked PVA hydrogel MRP.

Although PVA hydrogel MRP has good MR behaviour and mechanical properties, several issues need to be considered, such as desiccation when exposed to air and degradation caused by the oxidation of magnetic particles. In other words, the oxidation occurs due to the high utilisation of water as an aqueous swelling solution in the preparation of hydrogel [15,21,22,23]. These disadvantages limit the application of PVA hydrogel MRP, especially for long-term applications in the engineering field. Thereby, the utilisation of water during the preparation of the material needs to be superseded so that it can be used in long-term activities. Thus, having the ability to potentially choose an ideal solvent to tailor the properties of the materials could be an extremely effective approach to improve the disadvantages stated previously. It is well known that the good mechanical properties and behaviour of PVA hydrogel MRP are influenced by the type and composition of solvents used [24].

Previously, several studies reported and proved that in conventional PVA hydrogel (without CIPs), organic solvent such as dimethyl sulfoxide (DMSO) is compatible as dispersing phase in the fabrication of PVA hydrogel solutions [25,26,27,28]. Shi and Han [25] discovered that DMSO is another great alternative to be used as the solvent in preparing PVA solutions compared to water. Cha et al. [27] proved that the type of solvent and mixing ratio affects the structure and mechanical properties of the material. They found that the PVA hydrogels prepared from mixed ratio of DMSO/water have better mechanical properties compared to those of prepared using other solvents. This finding also agreed well with the results by Gupta et al. [28], which prepared PVA hydrogels in the mixture of DMSO and water. In the study, the PVA hydrogels show interesting characteristics depending on the ratio of the DMSO and water, which lead to better mechanical properties compared to those prepared using other solvents.

Even though the utilisation of DMSO/water in the preparation of conventional PVA hydrogel have been widely investigated, to the best of our knowledge, many current research have yet to investigate the influence of different solvent in chemically cross-linked hydrogel MRP on its dynamic viscoelastic properties, particularly the storage modulus and damping properties. Considering the impact of solvent on the mechanical properties of conventional PVA hydrogel, it is expected that the type of solvents would greatly affect the dynamic properties of chemically cross-linked hydrogel MRP. Furthermore, by changing the solvents, the corrosion and desiccation problems are expected to diminish so that the application of PVA hydrogels MRP can be extended. Therefore, the main technical contribution of this paper is to experimentally investigate the influence of different solvents on the dynamic viscoelastic properties of chemically cross-linked hydrogel MRP. Three different solvents: water, DMSO, and their binary mixtures (DMSO/water) with different weight fractions of CIPs were used as experimental parameter and the dynamic properties of hydrogel MRP were obtained under oscillation mode test using a commercial rheometer. Hence, the influences of applied magnetic field and strain amplitude were investigated thoroughly. Additionally, the stability and corrosion of the as-prepared PVA hydrogel MRP were observed in a month, and the degradation of the samples are presented and discussed accordingly.

## 2. Results and Discussions

### 2.1. Materials Characterisation

#### 2.1.1. FTIR Measurement

Figure 1 shows the FTIR spectra of the MRP samples with various solvents for samples S1-70, S2-70, and S3-70. The results are restricted to the MRP samples with CIPs concentration of 70 wt.%.

Figure 1 shows that five characteristics regions with strong transmittance peaks were observed at around 3300, 1600, 1400, 1004, and 948 cm^−1^. The broad peak at around 3000 to 3500 cm^−1^ can be assigned to the hydrogen bond, O–H stretching vibration region due to the intramolecular and intermolecular hydrogen bonds from the hydroxyl groups of PVA. The peak height at 3290 cm^−1^ for sample S1-70 increased compared to that of samples S2-70 and S3-70 due to the increase in O-H groups [26]. Moreover, the existence of peak at 3290 cm^−1^ belongs to the complexation of hydrogen bonding of OH groups from PVA borate ion as shown in Schemes 1 and 2. The result reveals that the PVA-borax having cross-linking network due to the complexation of PVA and borate ions. The peaks located at 1627 and 1404 cm^−1^ were associated with C=O stretching and C–O bending present in acetate groups, respectively. Meanwhile, the peaks at 1004 and 948 cm^−1^ can be assigned to the stretching vibration of S=O for DMSO that only existed in Samples 2 and 3. As for Sample 3, the intensity of those two peaks (S=O stretching) was reduced due to 20wt. % of water.

#### 2.1.2. Vibrating Sample Magnetometer

The magnetisation characterisation of the PVA hydrogel MRP with different solvents using a vibrating sample magnetometer (VSM) was successfully conducted as presented in Figure 2.

Figure 2 shows that all the MRP samples exhibit hysteresis loops of typical soft magnetic material behaviour. Figure 2a displays the hysteresis loops of hydrogel MRP (PVA/DMSO) with different CIPs contents. The *M_s_* for hydrogel MRP sample (PVA/DMSO) with CIPs of 50, 60, and 70 wt.% was 44.45, 60.69, and 74.67 Am^2^/kg, respectively. The increase in *M_s_* with the increase in CIPs content is a common phenomenon in magnetic materials. The higher the *M_s_*, the larger the responsive storage modulus (*G*′) of the samples towards the external stimuli, particularly magnetic field. Whilst, Figure 2b presents that the hysteresis loops of PVA hydrogel MRP with different solvents and fixed CIPs amount of 70 wt.%. The MRP samples display an unappreciable magnetic hysteresis as a prove that the type of solvent did not significantly change the magnetic saturation (*M_s_*). The *M_s_* for Samples S1-70, S2-70, and S3-70 was 76.21, 74.67, and 70.92 Am^2^/kg, respectively. The small difference in the *M_s_* value is due to the presence of different organic compounds in each of the samples that influence the reduction of the magnetic moment in CIPs inside each of the samples [2].

### 2.2. Rheological Properties

#### 2.2.1. Relative MR effect

The MR effect is known as the magnetic field-dependent property and the key parameter to investigate the performance of MR material. MR effects are mainly originated from the interaction of magnetic particles or in this case, the CIPs, when the MRP samples are exposed to the magnetic field. The CIPs are magnetised and tend to form chain-like structures with the same magnetic field direction [29,30]. The field-induced magnetic forces between the particles restrict the deformation of polymer chains, resulting in the changes of *G*′. Therefore, the reliance on the composition of CIPs and the type of solvents towards the dynamic mechanical properties of MRP was examined experimentally. Three MRP samples with various compositions of CIPs were prepared as shown in Table 1. The influence of magnetic flux density on shear storage modulus is presented in Figure 3. All MRP samples were tested at a fixed frequency of 1 Hz and strain amplitude of 0.01%. The applied external magnetic flux density was varied from 0 to 900 mT. To ensure the data consistency, all tests were run for three times, and the average values for both absolute and relative MR effect were calculated and presented in Table 1. The absolute MR effect is described as follow:(1)$$Absolute\;MR\;effect,\;\Delta G^{\prime }=G^{\prime }{}_{max}-G^{\prime }{}_{0}$$

While relative MR effect is the ratio of the absolute MR effect to the zero-field modulus, *G*_0_ and usually expressed in percent’s as below:(2)$$Relative\;MR\;effect=\frac{G_{max}-G_{0}}{G_{0}}\;\times \;100\%$$
where *G*_0_ is the zero-field modulus, and *G_max_* is the maximum modulus achieved in the presence of magnetic field.

Figure 3 shows that the hydrogel MRP samples exhibit MR effect behaviour, where the *G*′ increased consistently with the increase in magnetic flux density. A steep increase in *G*′ appears for all samples when the magnetic flux density increased from 0 to 900 mT. However, Figure 3a shows that the increase in *G*′ for Sample 1 was more obvious compared to that of the Samples 2 and 3 in Figure 3b,c. For examples, the *G*′ of samples with CIPs concentration of 70 wt.% (S1-70), displays the highest increase of 2.14 MPa at 900 mT. Meanwhile for Samples S2-70 and S3-70, the *G*′ increased up to 1.79 and 1.84 MPa, respectively. Besides, the saturated *G*′ is highly dependent on the composition of CIPs and significantly improved the MR effect. From Figure 3, the increment of *G*′ for all samples with a higher concentration of CIPs was more obvious compared to that of the samples with smaller concentrations. From these results, the average absolute and relative MR effect for all samples were calculated and summarised in Table 1. The relative MR effect of each sample can be calculated using.

From Table 1, the PVA hydrogel MRP with profoundly filled CIPs accomplished the highest relative MR effect. For example, the MR effect of S1-50, S1-60, and S1-70 was 5631, 11,540, and 15,544%, respectively, of which the MR effect increased with increasing of CIPs percentage. On the other hand, the maximum relative MR effect of samples with the highest concentrations of CIPs: S1-70, S2-70, and S3-70 increased to 15,544, 7049, and 11,024%, respectively. These findings agree with the results by Wang et al. [31], which found that the concentration of magnetic particles effectively influence the MR effect of the MR materials.

Table 1 also shows that different solvents affect the MR effect of the hydrogel MRP. The MR effect for Sample 2 was always the lowest compared to that of Samples 1 and 3 even though the concentration of CIPs was increased. In contrast, Sample 1 especially Sample S1-70 shows the highest MR effect among the samples due to the highest hydroxyl group caused by the highest amount of water content that soften the matrix. Thus, the CIPs were easily moved in Sample S1-70 to form the chain-like structures that restrict the polymer chains of PVA when subjected to the magnetic field. Sample S2-70 has a lower MR effect compared to Sample S3-70 because the utilisation of DMSO as the whole percentage of solvent contributed to the stiffness of the matrix. Additionally, based on Table 1, one can see that CIPs percentages and solvents type, with *p*-values less than 0.05, significantly affect the performance of PVA hydrogel MRPs toward MR effect. According to Krumova et al. [32], the PVA hydroxyl groups mainly consist of the hydrogen bonds that provide a major contribution towards the stiffness of the polymer chains. By introducing the DMSO as the solvent, the hydrogen bonding interaction diminished due to the existence of covalent bonding from sulphur and oxygen bonding resulting in the increment of the matrix stiffness. However, the stiffness of the matrix can be reduced by introducing the binary mixtures of DMSO/water as exhibited in Sample S3-70.

#### 2.2.2. Strain Amplitude Sweep

To understand the influence of different solvents on the viscoelastic properties of PVA-based hydrogel MRPs, strain amplitude tests under oscillatory shear rheometry were performed to determine the linear viscoelastic range of the MRPs. The LVE range is very important for viscoelastic materials like MRP to further comprehend the relationship between the microstructure rupture of MRP induced by external stimuli such as strain amplitude. Figure 4 shows the strain amplitude dependence of MRP with different solvent types on the *G*′ under 0 and magnetic field of 600 mT. The *G*′ is an important parameter to characterise the ability of the MRP to store energy after shear deformation.

Figure 4 shows that at the off-state condition (0 mT), the *G*′ remained unchanged with the increment of strain amplitude. All MRP samples approximately behave as linear viscoelastic materials as the *G*′ is independent of the strain amplitude at the off-state condition. Interestingly, at this condition, the *G*′ of MRP samples (see Figure 4a–c), was not increased even when many CIPs were embedded in the matrix. In other words, CIPs in the MRP without magnetic field were randomly distributed in the polymer matrix. A similar trend was also observed in the study reported by Mitsumata et al. [22]. At the on-state condition (600 mT), the *G*′ increased remarkably with the CIPs content for all MRP samples as shown in Figure 4a–c. The *G_max_* achieved by Samples S1-70, S2-70, and S3-70 was 2.370, 1.239, and 1.576 MPa, respectively. The increment of *G*′ indicates that in the presence of a magnetic field, the CIPs strengthen the matrix by reducing the distance between the particles. The strong dipole–dipole interaction between the CIPs is due to the magnetisation of CIPs to form chain-like structures attracted to each other inside the polymer chains.

Table 2 depicts the zero-field modulus (G′_0_), maximum modulus G′_max_, and magneto-induced modulus ∆G′ of all samples with 70 wt.% of CIPs. Sample S1-70 has the highest ∆G′ of 2.35 MPa, followed by Samples S3-70 and S2-70. It could be explained by the formation of hydroxyl group inside the matrix. The strong evidence of this statement is supported by the FTIR results in Figure 1. Due to the highest amount of hydroxyl group, the CIPs inside the matrix are easy to move, magnetise, and form chain-like structures when subjected to the magnetic field, therefore dramatically increased the ∆G′. Meanwhile, Sample S2-70 has the lowest ∆G′ of 1.21 MPa because it was made up of DMSO that increased the rigidity of the sample and reduced the elasticity of the MRP. Then, in Sample S3-70 (the binary mixtures of DMSO/water in a ratio of 80:20), the rigidity of the sample was reduced and the ∆G′ was improved.

Additionally, at the on-state condition, the *G*′ of the MRP samples decreased sharply with the increasing of strain amplitude. The sharp decrement of *G*′ was more pronounced for the MRP sample with 70 wt.% of CIPs when the strain exceeded 0.03%. This phenomenon is common in MR materials and called as a Payne effect. Generally, the Payne effect is the destruction or breakage of the magnetic particle chains due to the large amplitude strain and depends much on the CIPs content in the matrix [33]. Below critical strain ($\gamma_{c}=0.03\%)$, the material is still in within the LVE range where the materials are stable, and the chain structures have not been destroyed. Correspondingly, the range above $\gamma_{c}$ is known as non-linear LVE range, where the disruption of chains starts to occur. Figure 4 shows that the sort of solvents has almost no significant effect on the LVE range of MRP, however, the LVE can be enormously changed by the content of CIPs. Thus, from the results, the LVE range for MRP samples with various kind of solvents can be regarded as 0–0.03%.

#### 2.2.3. Damping Properties

The matrix and composition of CIPs also play a crucial role in the determination of damping properties of MRP. Damping properties reflect the capability of the material to dissipate energy in a vibrating system known as loss factor. The shear strain dependency of the loss factor is displayed in Figure 5.

From Figure 5, the loss factor for all MRP samples increased with increasing strain amplitude up to 100%. The off-state loss factor was higher than the on-state loss factor and shows linear variation with the strain amplitude. Generally, the dissipation of energy is caused by the interfacial slipping within the magnetic particles and the movement of polymer chains in the polymer [34]. However, in the absence of magnetic fields, the damping properties of materials is hinged on the polymer chains of the matrix. As noticed, Sample 1 shows the highest off-state loss factor at ~1.2% when the CIPs content was 50 wt.%. The astounding loss factor of Sample 1 compared to Samples 2 and 3 may be due to the liquid-like nature of the sample. The liquid-like nature of Sample 1 may be attributed to the higher formation of hydrogen bonds between the -OH groups of PVA, which will cause the polymer chains (soft segments) move more easily without being blocked in the matrix. Therefore, the loss factor was larger for Sample 1 compared to that of Samples 2 and 3 dues to the most minimal formation of hydrogen bonds. Meanwhile, Samples 2 and 3 can be considered as solid-like MRP because they comprise more hard segments compared to soft segments. The off-state loss factor of solid-like MRPs is lower because the interfacial slipping between the polymer chains barely happens due to stronger interactions between the polymer chains [35].

Upon the application of magnetic field, the behaviour of the on-state loss factor was obviously different from the off-state loss factor. Instead of long plateau up to the higher strain of ~10%, the on-state loss factor shows interesting behaviour. As illustrated in Figure 5a–c, the on-state loss factor was always lower than the off-state factor for all MRP samples. The on-state loss factor of MRP samples started to level off at very small strain up to 0.1% and increased with increasing strain amplitude. Interestingly, at the strain of greater than 0.1%, the plateaued region started to increase again. The increased loss factor can be related to the strain hardening of polymer chains as the strain increased. Strain hardening is a strengthening behaviour of the material during a large strain deformation caused by the large-scale orientation of polymer chains molecules. The increased loss factor at the higher strain caused by strain hardening is due to the rupture of polymer chains that increased the dissipation energy. This phenomenon was observed in plastic materials as reported in previous research [36]. Sample 1 shows larger strain hardening compared to Samples 2 and 3 because the liquid-like nature of the matrix caused more dislocation of the polymer chains, thus increased the loss factor.

Furthermore, Figure 5 shows that the loss factor decreased in the presence of the magnetic field; therefore, it is proven that less heat dissipation occurred during deformation due to stronger particle interactions. This is because, in the presence of magnetic field, the CIPs are magnetised and the magnetic interaction between the particles is increased. Thus, the interfacial slipping within the particles are reduced. Basically, the on-state loss factor is depending on the interaction between the magnetic particles. Therefore, with the presence of magnetic field, the magnetic force between the CIPs increases, thereby reducing the distance between CIPs which results in the decrement of interfacial slipping. Moreover, the loss factor of MRP samples also decreased with increasing CIPs concentration either in the off-state or the on-state conditions.

### 2.3. Desiccation and Corrosion Observation

Material stability is also an important criterion for assessing the performance of MR materials. As previously mentioned, hydrogel MRPs have a higher possibility for the particles to sediment or corrode over time. To prove the desiccation event occurred in the samples, three Samples—S1-70, S2-70, and S3-70—were cut into spherical disk with 20 mm in diameter and left exposed to the air for 24 h. The photograph of all samples is displayed in Figure 6.

Figure 6a shows that Sample S1-70 was desiccated and lost its softness (solidified) after exposed to open air for 24 h. The desiccation occurred because water starts to evaporate as the polymer structures are chemically less stable [37]. However, in Samples S2-70 and S3-70, the softness remained after 24 h of exposure to air as shown in Figure 6 b,c, respectively. It can be concluded that, with the addition of DMSO as a solvent during the dispersing phase, the solidification problems can be reduced. Other serious problems that mentioned previously, is the degradation of the samples due to the oxidation of CIPs that lead to corrosion and cause limitation in practical use. The photograph in Table 3 shows the degradation of the hydrogel MRP samples after a month due to the oxidation of CIPs that lead to corrosion problem.

Table 3 shows that Sample 1 (S1-50, S1-60, and S1-70) formed a yellowish layer due to the oxidation of CIPs. Furthermore, the yellowish layer was higher for the sample with the lowest content of CIPs (S1-50). Samples S1-50, S1-60, and S1-70 tended to corrode due to the existence of water molecules that contribute to the oxidation process of the CIPs. For Sample 2 (S2-50, S2-60, and S2-70), no yellowish layer was formed after a month, however, the CIPs settling was obvious for Sample S2-50. The oxidation of CIPs did not occur in Sample 2 as the samples were prepared with the DMSO without the existence of water molecules. Sample 3 (S3-50, S3-60, and S3-70) remained unchanged after a month. In fact, particles settling did not occur in Sample 3, which indicates that the stability of MRP is superior in the hydrogel MRP prepared from the binary mixtures of DMSO/water. The corrosion occurred in Sample 1 due to the higher water molecule compared to Samples 2 and 3. The numbers of water molecules inside the PVA solutions followed the order of, PVA hydrogel (water) > PVA hydrogel (DMSO/water) > PVA hydrogel (DMSO). This result shows a good agreement with the results obtained using molecular dynamic simulations studied by Shi et al. [25], which declared that water is the poorest solvent compared to DMSO for PVA hydrogel.

## 3. Materials and Methods

### 3.1. Material

In this study, the high molecular weight PVAs (≥98.0% hydrolysed, approximate molecular weight of 60,000) were purchased from Merck Company, Germany. Sodium tetraborate decahydrate (borax), 20 Mule Team Borax^TM^ was purchased from a drug store and used as a cross-linking agent. As a solvent, DMSO brand ChemAR was supplied by Systerm Chemicals. CIPs with a size of ~5 µm were procured by BASF (CC model). Deionised water was used to prepare the aqueous solution.

### 3.2. Preparation of PVA MRP

Firstly, a PVA solution of 7.5% (*w*/*v*) was prepared by dissolving 8.1 g of PVA beads in 100 mL of deionised water at 80 °C using a hotplate for 2 hr. Similar steps were repeated with a different mixture of organic solvent as illustrated in Table 4. The mixture was then cooled down to room temperature after completely dissolved. Secondly, the borax solution of 3% (*w*/*v*) was prepared by mixing the powder with deionised water and act as the cross-linking agent. CIPs with different concentrations of 50, 60, and 70 wt.% were added into each of PVA solutions. Next, the PVA solutions that contain CIPs were chemically cross-linked using borax solution with constant stirring using mechanical stirrer for 10 min. The mixture was kept overnight to obtain a uniform MRP sample.

### 3.3. Mechanism of Chemically Crosslinking

Unlike the physically cross-linked hydrogel MRPs, chemically cross-linked hydrogel MRPs arises from the presence of reversible chemical crosslinks of polyol-containing polymers that respond to the borate ion [38]. The chemistry applicable to the polyol-borate systems is shown in Figure 7. This interaction shows a di-diol complexation.

From Scheme 1 in Figure 7, the borate ions react with the diol unit of polymer to form mono diol. In Scheme 2, the monodiol reacts with a second diol unit of the polymer to form di-diol. A hydrogen bond is connected between the boron sites and PVA chains. The hydrogen bond is reversible and much lower than the chemical bond that leads to the self-healing ability of the materials after cleaving or shearing by force [38].

### 3.4. Characterization and Rheological Testing

The magnetic characteristics of MRP samples was evaluated using a VSM (Microsense 7404) at room temperature. The test was carried out using a broad range of magnetic field up to 1500 k/Am. The molecular structure of the PVA solutions was measured using FTIR (Bruker FTIR) spectrometer in the range of 4000–500 cm^−1^ at ambient temperature. Meanwhile, the dynamic mechanical properties of the hydrogel MRPs sample were measured using a rheometer (Model: MCR 302 Anton Paar) at controlled temperature of 25 °C. In this study, an oscillation mode was used to study the dynamic properties, strain amplitude, and magneto-field sweep test. The rheological testing for each sample has been repeated three times in order to reassure the consistency of the values. The mean, standard deviation, percentage of error, and the one-way ANOVA have been quantitatively calculated and presented. In one-way ANOVA analysis, the value of *p* = 0.05 was used to determine the variable (i.e., types of solvent) were statistically significant. Moreover, the stability test of MRP samples was thoroughly investigated and discussed in detail.

## 4. Conclusions

The present study was designed to determine the opportunities of using different solvents in the fabrication of PVA hydrogel MRP as an alternative to solve corrosion and desiccation problems as mentioned in previous studies. Three kinds of hydrogel PVA MRPs were prepared that contained water, DMSO, and their binary mixtures (DMSO/water) with ration of 80:20 by weight were prepared. Different percentages of CIPs (50, 60, and 70 wt.%) were added to each sample, making nine samples in total. The field-dependent rheological characteristics of all samples were experimentally investigated, and the corrosion event was observed for a month. The results show that the PVA hydrogel MRP prepared from water has the highest MR effect of 15,544% (Sample S1-70) with a CIPs concentration of 70 wt.%. Although Sample 1 has a higher MR effect than Samples 2 and 3, the presence of high-water molecules contributed to corrosion and desiccation problems. In contrast, Sample 2 (S2-70) and 3 (S3-70) have lower MR effects of 7049% and 11,024%, respectively, compared to Sample 1. However, Samples 2 and 3 have a better stability and no corrosion was observed after one month. In addition, the results show that the rheological properties of hydrogel PVA MRP samples are strongly influenced by the types of solvents and the CIPs concentration.

## Figures and Tables

**Figure 1 ijms-21-01793-f001:**
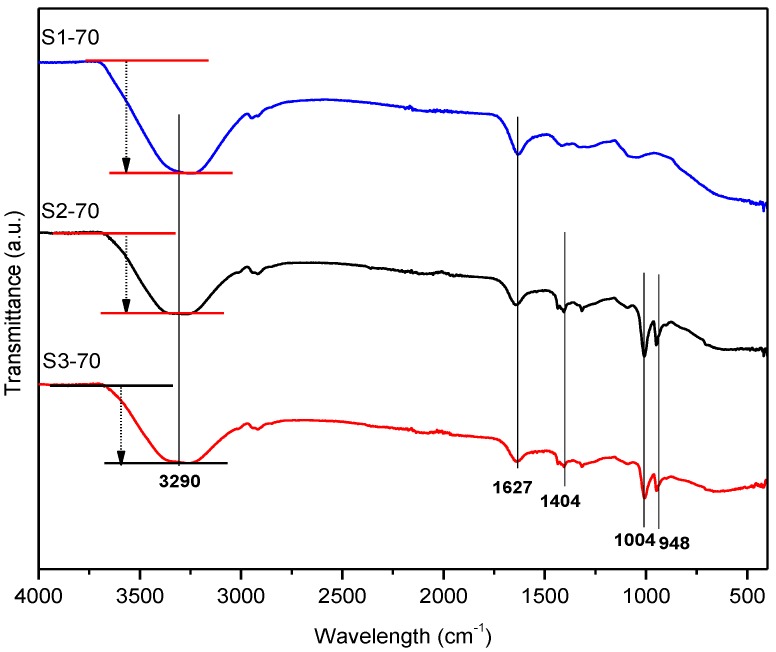
FTIR spectra of hydrogel magnetorheological plastomer (MRP) samples prepared with different types of solvents.

**Figure 2 ijms-21-01793-f002:**
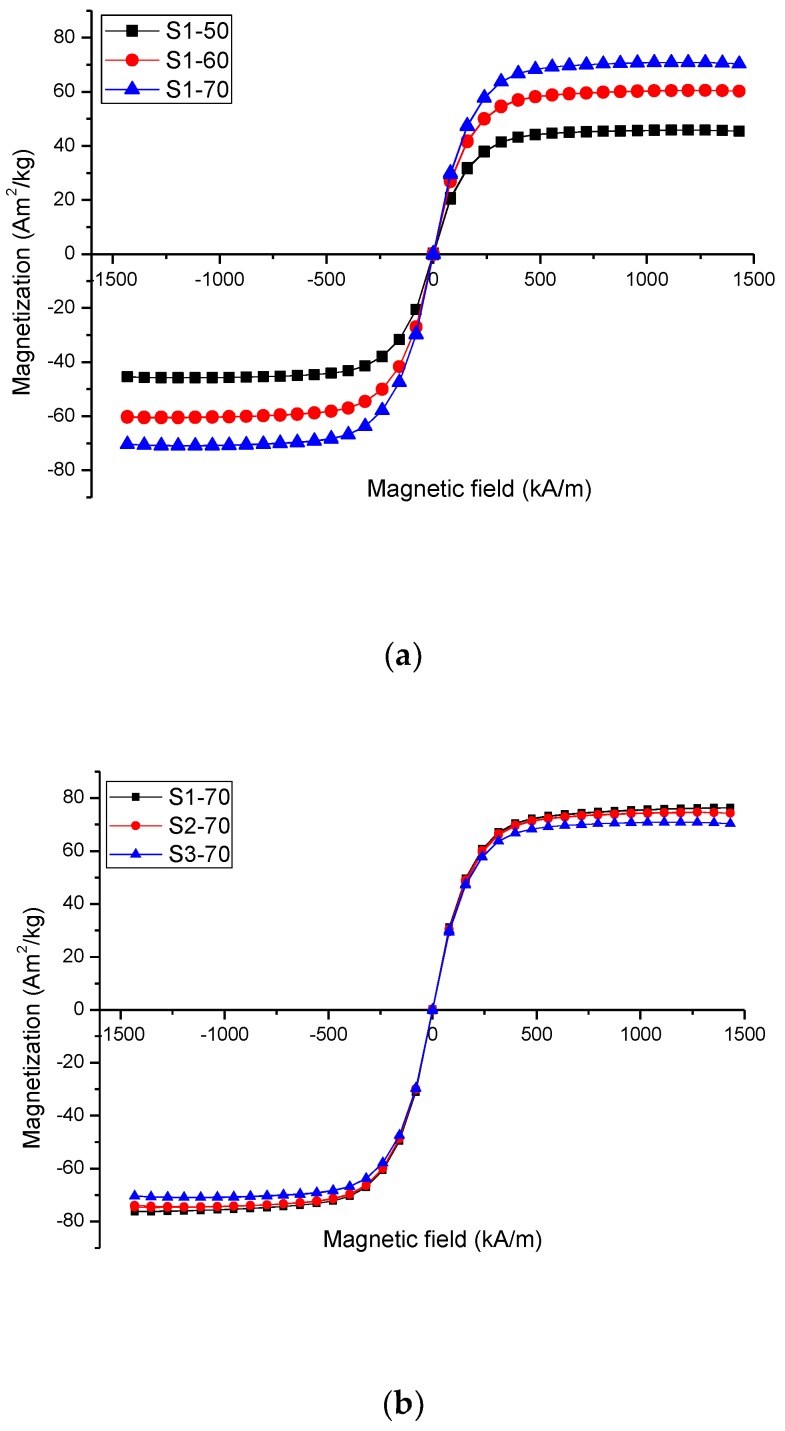
VSM profile of polyvinyl alcohol (PVA) hydrogel MRPs with (**a**) different carbonyl iron particles (CIPs) contents and (**b**) different types of solvent.

**Figure 3 ijms-21-01793-f003:**
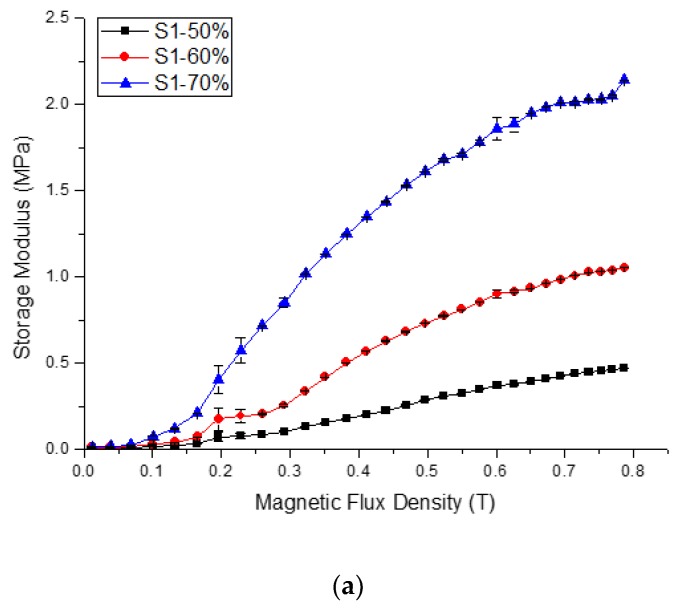
Shear storage modulus as a function of magnetic flux density with different CIPs compositions for (**a**) Sample 1, (**b**) Sample 2 and (**c**) Sample 3.

**Figure 4 ijms-21-01793-f004:**
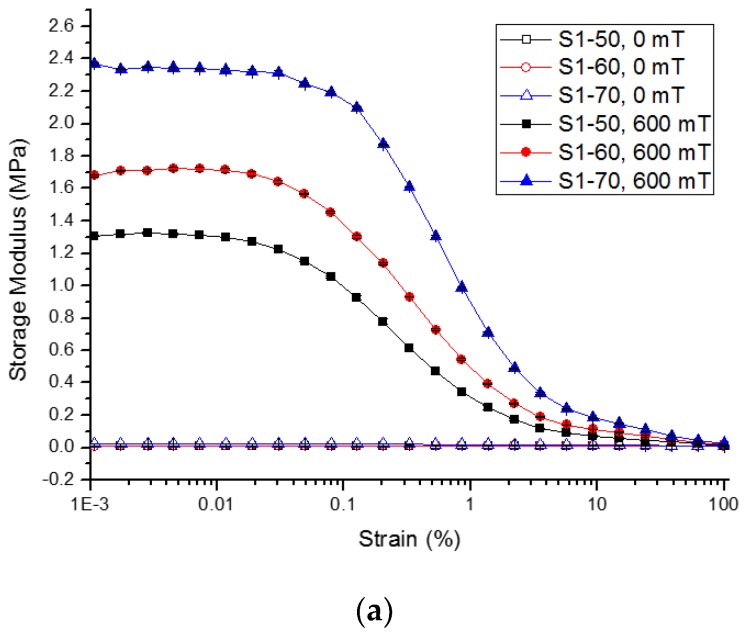
The dependencies of *G*′ (**a**) Sample 1 (**b**) Sample 2, and (**c**) Sample 3 as a function of strain amplitude. (*^at constant frequency, f = 1 Hz^).

**Figure 5 ijms-21-01793-f005:**
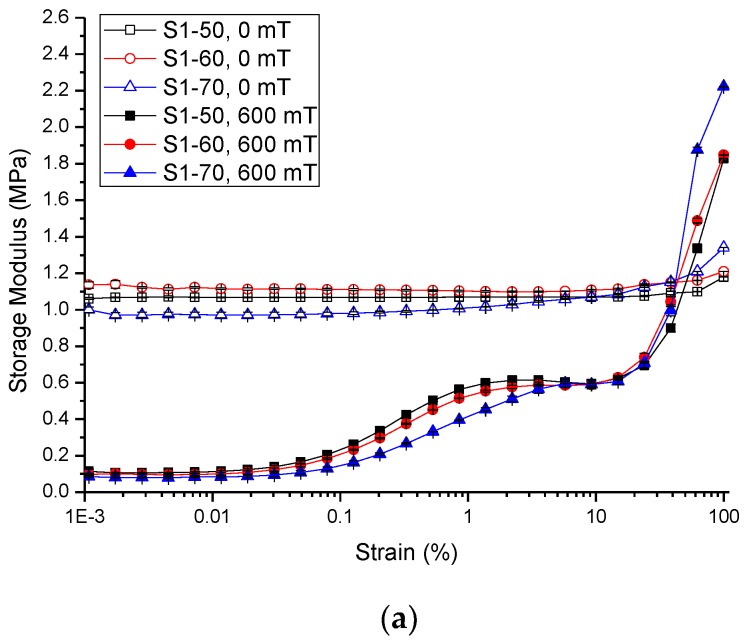
The dependencies of loss factor (**a**) Sample 1, (**b**) Sample 2 and (**c**) Sample 3 as a function of strain amplitude. (*^at constant frequency, f = 1 Hz^).

**Figure 6 ijms-21-01793-f006:**
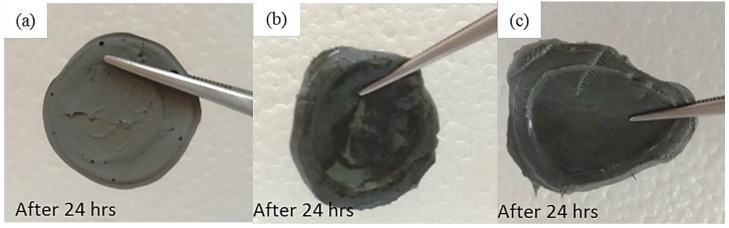
The desiccation observations after 24 h for sample: (**a**) S1-70, (**b**) S2-70, and (**c**) S3-70.

**Figure 7 ijms-21-01793-f007:**
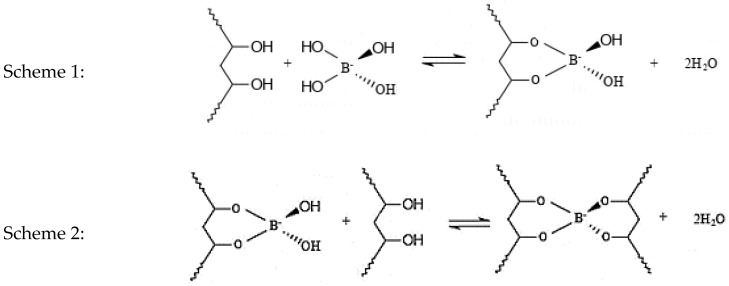
The cross-linking structure of compound formation between PVA hydrogel and borax.

**Table 1 ijms-21-01793-t001:** The absolute and relative magnetorheological (MR) effect hydrogel MRP with different solvents and CIPs content.

**Sample**	**Initial Storage Modulus, G′_0_**					**Max. Storage Modulus, G′_max_**					**Avg. Absolute MR Effect**	**Avg. Relative MR Effect**	
	**1**	**2**	**3**	**Mean**	**^a^ SD**	**1**	**2**	**3**	**Mean**	**^a^ SD**	**MPa**	**%**	**^b^*p***
S1-50	0.008	0.008	0.009	0.008	0	0.472	0.473	0.473	0.473	0	56.307	5631	0.002
S2-50	0.012	0.006	0.012	0.010	0.001	0.206	0.207	0.207	0.207	0	19.939	1994	
S3-50	0.008	0.008	0.008	0.008	0.001	0.421	0.422	0.421	0.421	0	50.231	5023	
S1-60	0.009	0.008	0.010	0.009	0.004	1.055	1.055	1.055	1.055	0	115.401	11,540	< 0.001
S2-60	0.025	0.021	0.024	0.023	0.002	0.570	0.572	0.570	0.571	0.001	23.442	2344	
S3-60	0.013	0.011	0.013	0.012	0.002	0.802	0.803	0.802	0.802	0	63.426	6343	
S1-70	0.013	0.015	0.013	0.014	0	2.140	2.140	2.140	2.140	0	155.448	15,544	< 0.001
S2-70	0.026	0.023	0.026	0.025	0.001	1.791	1.792	1.791	1.792	0.001	70.491	7049	
S3-70	0.017	0.016	0.017	0.017	0.001	1.841	1.841	1.841	1.841	0	110.242	11,024	

^a^ SD = standard deviation. ^b^ each sample *p* value was obtained through ANOVA, with variance of solvents effects towards MR effect.

**Table 2 ijms-21-01793-t002:** The zero-field modulus (G′_0_) maximum modulus (G_max_) and magneto-induced modulus ∆G′.

**Sample**	**Initial Storage Modulus, G′_0_**					**Max. Storage Modulus, G′_max_**					$\Delta G^{\prime }$
	**1**	**2**	**3**	**Mean**	**SD**	**1**	**2**	**3**	**Mean**	**SD**	**MPa**
S1-70	0.021	0.019	0.02	0.020	0.001	2.366	2.357	2.387	2.370	0.015	2.35
S2-70	0.032	0.036	0.034	0.034	0.002	1.236	1.236	1.246	1.239	0.006	1.21
S3-70	0.02	0.019	0.018	0.019	0.001	1.665	1.562	1.501	1.576	0.083	1.56

* SD = standard deviation.

**Table 3 ijms-21-01793-t003:** Photograph images of MRP samples conditions after a month: Sample 1, Sample 2, and Sample 3 with different concentrations of CIPs.

Sample	CIPs Concentration (After a Month)		
	50%	60%	70%
1	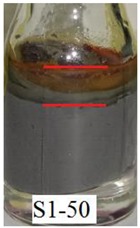	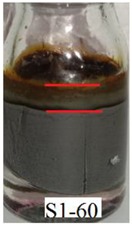	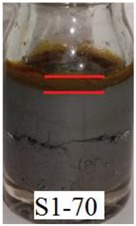
2	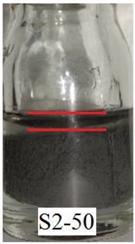	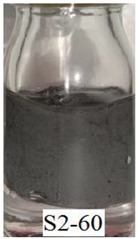	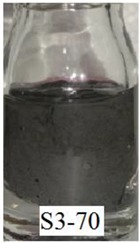
3	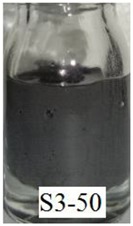	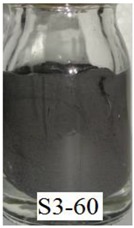	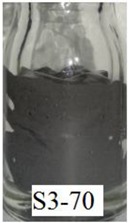

**Table 4 ijms-21-01793-t004:** The compositions of hydrogel MRP with different types of solvents.

	Sample	CIPs Content, wt.%	PVA Content, wt.%
**Sample 1 (PVA/water)**	S1-50	50	50
	S1-60	60	40
	S1-70	70	30
**Sample 2 (PVA/DMSO)**	S2-50	50	50
	S2-60	60	40
	S2-70	70	30
**Sample 3 (PVA/DMSO:water)**	S3-50	50	50
	S3-60	60	40
	S3-70	70	30

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
