# Peer review of "Solvent Dependence of the Rheological Properties in Hydrogel Magnetorheological Plastomer"

_ijms, 2020, doi:10.3390/ijms21051793_

Round 1
Reviewer 1 Report
The manuscript entitle “Solvent Dependence of the Rheological Properties in Hydrogel Magnetorheological Plastomer” provides an interesting study useful for PVA-based hydrogel preparation.
Several points must be addressed before being considered for publication:
1) The sentence “However, ….” in lines 16-19 is not written properly. Similar errors are in line 112 (be investigated) and line 118 (are also being)
2) The authors name a mixture of two solvents (DMSO and water) as a solvent. I think that it would be proper to call it mixture.
3) The information of the legend of Fig 2 is wrong. The same states in lines 152 and 153 when they talk about Figure 2(a).
4) All the results in this study (Graphs and tables) show values without indicated the mean value and SD values. Thus, no statistical analyses are performed in order to determine if the changes are statistically significant. The authors do not mention either how many times they have performed the experiments in order to check the reproducibility.
5) In line 372 the lack of a reference shows an error warning that must be corrected.
Author Response
Point 1: The sentence “However, ….” in lines, 16-19 is not written properly. Similar errors are in line 112 (be investigated) and line 118 (are also being)

Response 1: Thank you for the comment. The sentence has been corrected and updated in the text as suggested. (page 1, line 17-18)
Point 2: The authors name a mixture of two solvents (DMSO and water) as a solvent. I think that it would be proper to call it the mixture.
Response 2: Thank you for the comments. After considerations, the word solvent to address ‘a mixture of two solvents’ has been changed to ‘binary mixtures’ rather than ‘binary solvents’
(page 1, line 26)
Point 3: The information on the legend of Fig 2 is wrong. The same states in lines 152 and 153 when they talk about Figure 2(a).
Response 3: Thank you for the comments. All the information related to Figure 2 has been corrected including legendary information. (page 4, Line 146-157)
Point 4: All the results in this study (Graphs and tables) show values without indicated the mean value and SD values. Thus, no statistical analyses are performed in order to determine if the changes are statistically significant. The authors do not mention either how many times they have performed the experiments in order to check the reproducibility.
Response 4: Thank you for the comments. All graphs and data in the manuscript were updated and plotted using mean value. The average and standard deviation, SD had been analyzed. The changes in all data are proven significant by using one-way ANOVA. The ANOVA analysis is performed to prove that the types of solvent are significantly influenced the MR performance mainly the MR effect as the MR effect is the key factor to study the performance of MR materials. The ANOVA results are attached as a reference in Attachment 1. Based on the ANOVA results, the P-value is less than 0.05 which means that the hypothesis is significant. (page 5&8, 203-205)
Point 5: In line 372 the lack of a reference shows an error warning that must be corrected.
Response 5: Thank you for the comment. The error in the sentence has been corrected. (page 13, line 383)

Reviewer 2 Report
The current manuscript by Hapipi et al. represents an experimental investigation of the role of 2 solvents and their mixture on the magnetorheological properties of carbonyl iron – PVA gels. In general, the manuscript’s scientific contributions are clear, although there are a few minor issues to be addressed. The text quality, however, must be improved significantly prior to publication:
Scientific questions to be addressed:
Ln 169: You claim that the storage modulus increase is due to formation of chain-like structures. Is there explicit evidence for the formation of such structures (literature or your own observations)?
Table 1: What are the G0 values and their standard deviations. What is the typical error in percentage? If the values have up to 2 times difference, as in Table 2, it is perhaps more appropriate to remove the exact numbers and to put the orders of magnitude increase.
Examples of technical issues and language corrections:
1). The English language needs an extensive editing to make the manuscript more readable. For instance, rolls 209-212 constitute the following sentence:
“In contrast, Sample 1 especially sample, S1-70 showed the highest MR effect amongst samples might due to the highest hydroxyl group cause by highest amount of water content that soften the matrix.”
The sentence repeats the word “sample” 4 times and completely omits the existence of the the verb “to be”. Another example of sentences without verbs could be found just a couple of rows below:
According to Krumova et al. [32], the PVA hydroxyl groups mainly the hydrogen bonds that provided a major contribution towards the stiffness of the polymer chains.
2) There are plenty of technical mistakes. Just to name a few:
Table 1: “The absolute and the relative MR effect…” whereas in the table is given only the relative effect.
Ln. 254. the authors write ‘0f’ instead of “of”
Table 2: dG for S1-70 is calculated wrong: 2.73 in not equal to 2.395-0.021
Ln. 372: “Error! Reference source not found..” could be found instead of reference number
Ln. 373: “…was hen cooled”, I assume no chicken was involved in the cooling?
Ln. 422-427: References have no numbering
Author Response
"please see the attachment"

Round 2
Reviewer 1 Report
The authors have addressed well the reviewer's comments.